



# Coupling between the North Atlantic subpolar gyre vigor and forest fire activity in northern Scandinavia

Tine Nilsen[1,2], Dmitry V. Divine[3,1], Annika Hofgaard[4], Andreas Born[5,6], Johann Jungclaus[7], and Igor Drobyshev[8,9]

[1]Department of Mathematics and Statistics, UiT - The Arctic University of Norway, 9037 Tromsø, Norway
[2]Department of Geography, Justus-Liebig University of Giessen, 35390 Giessen, Germany
[3]Norwegian Polar Institute, Fram Centre, 9296 Tromsø, Norway
[4]Norwegian Institute for Nature Research, 7485 Trondheim, Norway
[5]Department of Earth Science, University of Bergen, 5020 Bergen, Norway
[6]Bjerknes Centre for Climate Research, Bergen, 5020 Bergen, Norway
[7]Max Planck Institute for Meteorology, 20146 Hamburg, Germany
[8]Southern Swedish Forest Research Centre, Swedish University of Agricultural Sciences, 230 53 Alnarp, Sweden
[9]Chaire industrielle CRSNG-UQAT-UQAM en aménagement forestier durable, Université du Québec en Abitibi-Témiscamingue (UQAT), J9X 5E4 Québec, Canada

**Correspondence:** Tine Nilsen, tine.nilsen@uit.no

**Abstract.** The circulation strength of the North Atlantic subpolar gyre varies on a range of timescales, it regulates the northwards oceanic heat transport and influences weather and climate over Scandinavia. We test the hypothesis that persistent atmospheric circulation anomalies favorable for extensive forest fire activity in the northern Scandinavian boreal region are predominantly associated with weaker subpolar gyre strength on subannual timescales. We included both winter and summer

5   drought as important precursors for forest fire occurrence in the boreal region. Three ensemble members of climate model simulations covering the time period 850-2005 AD are considered. Years of widespread and severe drought in northern Scandinavia are identified using the monthly drought code as a summer-drought indicator, and winter drought is identified as the 5th percentile of coldest and driest winters. The statistical significance of anomalous ocean- and atmospheric circulation is tested for these years, both during and prior to the main fire season.

10   Analysis of the ensemble of three simulations did not yield a generalized result, hence the hypothesis cannot be confirmed for subannual timescales. For the three simulations we find respectively that the fire-prone years are associated with subpolar gyre circulation that is on average stronger, weaker or insignificantly changed compared with the mean state. The 5th percentile of most extreme dry and cold winters has a strong relation to the winter North Atlantic oscillation (NAO), but not with the gyre circulation state. We find a number of extremely cold/dry winters occurring during the Little Ice Age (LIA, 16th-19th centuries

15   AD), and infer that winter drought may have played a significant role in promoting forest fire activity at this time.

Our results highlight the importance of resolving the past fire seasonality in the northern Scandinavian domain, and developing compound drought indicators for winter and spring.





# 1 Introduction

Forest fire activity is the main driver of ecosystem dynamics in the boreal forest. It is closely coupled to the atmospheric
circulation and its annual variability at the regional scales, controlled predominantly by climate (Swetnam, 1993; Westerling
et al., 2006). A better understanding of the natural driving processes for boreal forest fire activity may help improve forest fire
prediction systems, an important element of climate change adaptation strategies.

Persistent, dry weather conditions favorable for forest fire activity in Scandinavia are often associated with atmospheric
blocking events in the Northeast Atlantic (Brunner et al., 2017). Atmospheric blocking events are weather patterns with per-
sistent high-pressure systems that shift the westerly storm tracks over the affected region (Rex, 1950). Blocking events occur
year around in the North Atlantic realm, though they are more frequent during winter. They occur quasi-randomly and hence
feature weak predictability both in operational weather forecasts and in model simulations..

Atmospheric blockings may drive changes in ocean circulation, and correlate with sea surface temperatures (SST) (Häkkinen
et al., 2011). Correlations between SST and forest fire activity have been investigated and identified predominantly for North
America (Cook et al., 2007; Kitzberger et al., 2007; Kushnir et al., 2010), but also for Scandinavia (Drobyshev et al., 2016).

The North Atlantic subpolar gyre is a potential key region for improving predictive skill of forest fire activity because it is
a region with high predictive potential on interannual to decadal timescales (Msadek et al., 2010; Årthun et al., 2017; Brune
et al., 2018; Buckley et al., 2019; Kushnir et al., 2019). The variability on shorter timescales is controlled mainly by wind-
stress forcing (Böning et al., 2006; Born and Stocker, 2014). Multidecadal to centennial-scale variability of the subpolar gyre
circulation and its climatic effects have been demonstrated in proxy-based reconstructions (Moffa-Sánchez and Hall, 2017;
Thornalley et al., 2017), and in simulations of general circulation models (GCMs) (Born et al., 2009; Born and Mignot, 2012;
Born et al., 2013; Moreno-Chamarro et al., 2017). Weaker circulation is on average associated with reduced northwards heat
transport, stronger and more frequent blocking events in the Northeast Atlantic according to model and observation studies
covering the past decades to centuries (Langehaug et al., 2012; Moffa-Sánchez et al., 2014; Moreno-Chamarro et al., 2017).
Atmospheric variability triggers oceanic circulation changes in the subpolar gyre, but may also promote drought directly
(Barsugli and Battisti, 1998; Häkkinen and Rhines, 2004; Häkkinen et al., 2011; Drijfhout et al., 2013; Kleppin et al., 2015;
Duchez et al.).

In this study we test if the variability in the subpolar gyre circulation may act as an amplifier for the atmospheric blocking
activity over the Northeast Atlantic and Scandinavia, with implications for the emergence of climate conditions associated with
increased risk of forest fires. Specifically, we test the hypothesis that the most extensive and persistent droughts over northern
Scandinavia, defined here as the region north of 64°N, 12-28°E occur predominantly during periods of reduced circulation
strength of the North Atlantic subpolar gyre.

Past-millennium and historical climate model runs are used in this work. Our hypothesis testing is divided in two parts,
where part (**i**) comprises summer forest fires that occur due to warm and dry spring and summer conditions only. For the
first time we apply the monthly drought code index (MDC), (Girardin and Wotton, 2009), to identify model-years subject
to extensive summer drought and high temperatures. Potential large fire years in northern Scandinavia are expected to be





associated with weakened circulation of the subpolar gyre, positive anomalies in sea surface temperature (SST) over the same area and prevailing anomalies in sea level pressure and atmospheric geopotential height variables over Scandinavia.

Part (**ii**) involves partial attribution of boreal forest fire activity to cold-season drought, which the MDC cannot capture as it measures warm-season drought only. Winter droughts and -forest fires have been observed in Norway and Sweden in recent time, although with lower frequency than for summer fires. An example from western Norway is the winter 2013-2014 when a massive high-pressure system prevailed over Scandinavia for many weeks. Persistent dry air masses and easterly winds caused severe drought with numerous forest fires during the later part of January (Schmuck et al., 2015; Rasmijn et al., 2016).

We motivate the relevance of winter drought by the observation that a high number of forest fires are recorded for the Little Ice Age period (LIA, 16th-19th centuries), (Drobyshev et al., 2016). Proxy-based reconstructions and climate model studies show that the LIA was characterized by overall cold and dry conditions in Europe (Bradley and Jones, 1993; Moore et al., 2001; Mann et al., 2009; Seftigen et al., 2017). Recent climate field reconstructions of summer surface air temperatures (SAT) confirm low temperatures during the LIA for Scandinavia (Luterbacher et al., 2016; Werner et al., 2018). The effects of precipitation deficits and anomalous temperatures both contribute to drought in the northern Scandinavian region, although the former may play the dominant role (Drobyshev et al., 2016, supplementary material).

## 2 Data and methods

### 2.1 Data

We use monthly climate model data from the MPI-ESM-P millennium-long (past-1000-R1, -R2 and -R3) runs and associated historical runs (Jungclaus et al., 2014) for the time period 850-2005 AD. The concatenated data sets will in the following be denoted the R1, R2 and R3 simulations respectively. Considered atmospheric climate variables include maximum SAT, precipitation, sea level pressure and geopotential height at 500/300 hPa. The oceanic variables are the barotropic streamfunction as a metric for oceanic circulation vigor/strength of the subpolar gyre, SST and atmosphere-ocean heat flux (positive in the direction from the atmosphere to the ocean).

The MPI-ESM-P consists of the general circulation models for the atmosphere ECHAM6 and the ocean/sea-ice model MPIOM. ECHAM6 is run at T63 spectral resolution, MPIOM features a conformal mapping horizontal grid with resolution less than 20 km in the North Atlantic. The simulations over the last millennium follow the Paleoclimate Modeling Intercomparison Project, Phase 3 (PMIP3 - past1000) experimental protocol [Schmidt et al., 2011] and include estimates for the external drivers volcanic aerosols, solar insolation variations, greenhouse gases, and land-cover changes.

For hypothesis part (**i**), the anomalies of the barotropic streamfunction, SST, atmosphere-ocean heat flux, sea level pressure and the geopotential height at 500/300 hPa are studied for the months preceding and during exceptionally dry and warm summers as indicated by seasonal monthly drought code values. Part (**ii**) follows a similar scheme, but for the most extreme cold and dry winters. Here, we utilize directly the index of atmospheric blocking events estimated in Moreno-Chamarro et al. (2017) for the R2 simulation, following Scherrer et al. (2006) for inference on winter atmospheric circulation anomalies.



## 2.2 Methods

### Identifying potential large fire years using the monthly drought code

The monthly drought code (MDC) is a soil moisture index, calculated cumulatively from monthly maximum SAT and monthly precipitation for the warm season (April to October) (Girardin and Wotton, 2009). The index is used here as a measure for summer drought conditions, and is calculated for 22 model grid cells in the northern Scandinavian region for the period 850-2005 AD. For each grid cell in the target region, the maximum MDC values are typically observed during July or August, and are generally similar to those estimated for Canada (Girardin and Wotton, 2009). The MDC was originally designed and implemented for Canadian soils, but is calculated for Scandinavia in Drobyshev et al. (2012).

The observed frequency of real-world large fire years (LFY) during the period 1273-1933 AD is approximately one or two events per century with a few exceptions (Drobyshev et al., 2015, conservative classification protocol). Between 20-50% of sites burned during a single large fire year, (Drobyshev et al., 2015, supplementary material). We set a threshold for the MDC defining a potential large fire year in the model simulations, to identify model-years with similar frequency and spatial extent of extreme drought as the true large fire years. The potential LFY is defined as a model year with more than 30% of the northern Scandinavian grid cells exceeding a July MDC value of 320. Girardin and Wotton (2009) set the limit of extreme drought to MDC>280, our stricter choice is a necessary adjustment to obtain the desired frequency and number of potential large fire years over the analysis period.

### Winter drought analysis

Another measure is needed to capture the cold-season drought, and without any indices comparable to the warm-season MDC we choose as a simplistic approach to extract the 5% simulated climate model years with simultaneous driest and coldest winters over northern Scandinavia (December, January, February, DJF). Spatially averaged SAT and precipitation values are estimated for the northern Scandinavian domain for the winter season (DJF) between AD 850-2005. Extreme winter drought and cold anomalies are identified as the lowest 5th percentile of the respective precipitation and SAT anomalies. These are considered possible contributors to forest fire activity.

### Calculation of anomaly composites and hypothesis testing

Monthly and seasonal anomalies of climate model variables are calculated for every identified fire-prone year, and every grid cell in the region of interest (region varying according to Figs. 1 and 4). For each set of anomalous years (see Table A1), composites were constructed by averaging in time. Statistical significance testing based on bootstrapping is applied for these anomaly composites (von Storch and Zwiers, 1999). Bootstrapping involves resampling from the original data that have the same distribution as the anomalous data to test. From the 1156 available model years of each simulation, resampling sets are drawn with the same size as the anomalous data. Anomaly composites are calculated, using a total of 5000 bootstrapping experiments. The 95% confidence range is based on the distribution of representative composites.





**Winter subpolar gyre index**

An area-averaged index is estimated for the strength of the winter subpolar gyre circulation by calculating the spatial mean
of the barotropic stream function for DJF over the region $50° − 65°$N and $60° − 10°$W, multiplying by -1 to account for the
generally cyclonic gyre circulation. The standardized time series is shown in Fig. S1, calculated by subtracting the long-term
mean from the annual values. On a year-to-year basis we test whether the 5% most exceptional dry and cold winters have a
negative or positive subpolar gyre index value, indicating a weak or strong gyre circulation, respectively. This method repre-
sents a time-resolving alternative to analyzing the anomaly composites described above, which are averaged in time.


## 3  Results

All individual years identified either as warm and dry using the MDC threshold or as cold and dry using the 5th percentile of
SAT and precipitation are listed in Tab. A1.

### 3.1  Oceanic and atmospheric variability prior to and during potential large fire years using the MDC threshold

The composite anomalies for the R1, R2 and R3 simulations over selected seasons are calculated (Figs. 1, 2 and 3). The model
variable anomalies include, respectively, the DJF barotropic streamfunction, atmosphere-ocean heat flux and SST (panels a-c),
preceding dry and warm summers. Sea level pressure for April-August (AMJJA), or June-August (JJA) during the warm and
dry summers is also considered (panel d).

All three model simulations are characterized by an anomalous loss of latent and sensible heat from the subpolar gyre
preceding dry and warm summers. We find robust anticyclonic atmospheric circulation anomalies over northern Scandinavia
during the summer months, and in two out of three simulations these anomalies persist between April and August (Figs. 1d
and 2d). From the definition of the MDC the anomalous years also exhibit positive anomalies in summer maximum SAT, and
precipitation deficits over northern Scandinavia (figures not shown). Anomalous surface summer warming in the Norwegian
Sea is also observed.

For the barotropic streamfunction and SST anomalies in preceding winters, results deviate between the simulations:

**R1:** The barotropic streamfunction composite anomalies in the western subpolar gyre are weakly negative (insignificant),
indicating stronger gyre circulation in the area of Labrador Sea water formation (Figs. 1a,c). SST anomalies are also negative.

**R2:** The anomalies in barotropic streamfunction and SST are significantly positive as expected for weaker cyclonic circula-
tion in the western subpolar gyre (Fig. 2a,c). Similar but weaker anomalies are observed also during spring and summer (Fig.
S2). Based on the positive anomalies in the 300 hPa geopotential height during February and March we infer a weak excitation
of the tropospheric polar vortex (Fig. S3).

**R3:** The MDC threshold is adjusted for the R3 run from 320 to 280 to mimic the frequency of potential large fire years similar
to observations. Model-specific sensitivity is the most likely cause of discrepancies in summer drought conditions between the
model simulations; the infinitesimal differences in initial conditions leads to different average climates. For instance, a bias in





the position of the atmospheric jet stream associated with sea ice extent anomalies could influence the average climate over Scandinavia.

Composite streamfunction anomalies are insignificant in the subpolar gyre (Fig. 3a). As for the R1 run we find negative anomalies in atmosphere-ocean heat flux and SST (Fig. 3b-c).

## 3.2 Oceanic and atmospheric variability during extreme cold and dry winters

Between 0-5 years per century prone to extreme winter drought are identified over the simulated time period. The occurrence of winter drought is particularly high during the LIA (see Tab. A1), making the total distribution of model fire-prone years more similar to observations.

For the composites of winter drought, we find respectively that around 65%, 71% and 50% of the coldest and driest winters
occur when the gyre circulation is weak according to the sign of the subpolar gyre index (Fig. S1, 4a, 5a and 6a). The SAT and the geopotential height at 500 hPa display similar characteristic dipole patterns for all simulations. However, the remaining composite anomalies are common only for R2 and R3.

**R1:** There are no significant heat exchange anomalies between the atmosphere and the ocean in the subpolar North Atlantic (Fig. 4c). Barotropic streamfunction composite anomalies are positive in the western subpolar gyre, but SST anomalies are
insignificant.

**R2 and R3:** These simulations exhibit a winter atmospheric circulation state with positive 500 hPa geopotential height anomalies in the northwestern subpolar North Atlantic including Greenland and negative anomalies over the North Atlantic within the 40-60°N latitude (Fig. 5b and 6b). Accordingly, the average DJF atmospheric blocking index for R2 shows an increased frequency of blocking over Greenland and Scandinavia for these years, with more than 40 and 20 days under blocked
conditions over Greenland and Scandinavia, respectively (Fig. 5c). The SAT anomalies suggest anomalously cold conditions over Scandinavia and a warming in the Greenland region (Fig. 5b and 6b). In general, the atmospheric circulation state reveal a pattern typical of the negative phase of the North Atlantic oscillation (NAO) during cold and dry winters. All the coldest and driest winters except one are associated with the negative NAO index values.

Diminished ocean heat loss is observed as positive heat flux anomalies at the ocean surface between 45-60°N north in the
subpolar gyre. The western part of the subpolar gyre is likely strengthened as the barotropic stream function anomalies are negative, with a tripole pattern in the North Atlantic SST (Fig. 5d and 6d). The ocean circulation pattern emerging between the subpolar and subtropical gyres correspond to the so-called intergyre gyre (Marshall et al., 2001b, a; Núñez Riboni et al., 2012), defined as the leading empirical orthogonal function (EOF) of the sea surface height (Núñez Riboni et al., 2012). The oceanic circulation pattern is in turn coupled to decadal-to-multidecadal variability in the NAO.



## 4 Discussion

Our proposed mechanistic coupling between the circulation strength of the subpolar gyre and summer drought emergence in northern Scandinavia is clearly observed in the R2 model simulation for the MPI-ESM-P, while the analyses of the R1 and R3 simulations do not yield conclusive results. The winter droughts appear to be closer related to the state of the NAO than to the subpolar gyre vigor. The inconclusive results of our study are of value since the ocean-atmosphere coupling is intrinsically difficult to describe and model on subannual timescales. Analysis of the subpolar gyre vigor could potentially lead to a better physical understanding of the reported correlation between North Atlantic SST and Scandinavian forest fire activity from Drobyshev et al. (2016). However, our results show that there is no direct correspondence between the subpolar gyre vigor and atmospheric blocking activity down to the subannual timescale, at least in the model world. The focus on short timescales, along with the prerequisite that MDC-based drought is used to indirectly infer about atmospheric blocking are the two constraints of our study which makes it unique compared with previous model-studies of the gyre-blocking coupling in the subpolar North Atlantic. Our results also highlight the limited ability of commonly used fire weather proxies to capture the complexity of the weather conditions contingent with increased fire activity.

Weaker subpolar gyre cyclonic circulation during winter is likely contributing to the heat accumulation in the northern North Atlantic region observed for the R2 simulation in model-years prone to summer forest fire activity. Accompanied with anomalous heat loss to the atmosphere (Fig. 2b), this pattern may alone be sufficient to trigger and sustain the persistent atmospheric circulation anomalies over northern Scandinavia from spring to late summer. However, considering also the R1 and R3 simulations we demonstrate that this is not the only possible pattern of oceanic variability leading to seasonal scale regional circulation anomalies triggering summer drought.

The consistent oceanic heat loss observed prior to dry summers is associated with different patterns of SST anomalies in the three simulations, indicating a disconnection between the atmospheric and oceanic states. Atmospheric blocking frequency in the North Atlantic European region is generally underestimated for GCMs used for CMIP5 experiments, and higher atmospheric resolution is necessary to achieve similar blocking frequencies as observed for reanalysis data (Müller et al., 2018, and references therein). Furthermore, the lack of consensus between the simulations in our study might indicate model uncertainties for the ocean-atmosphere coupling on the subannual timescales for the MPI-ESM-P.

Similar analysis based on observational data is substantially hampered by a generally low frequency of years subject to extreme drought, and hence a limited number of cases available, with only 5 years classified as LFY for the reanalysis period (Drobyshev et al., 2016). Random variability inherent to the seasonal climate may mask the drought signal. Moreover, establishing a link between the subpolar gyre circulation anomalies and Scandinavian droughts in the instrumental data is further hampered by a general inconsistency between the ocean reanalysis data products for the subpolar gyre region prior to the 1980s (Born et al., 2015). However, our results demonstrate some agreement with observational data-based studies. In particular, Drobyshev et al. (2016) in their analysis of instrumental SST find winter-to-spring negative SST anomalies in the subpolar gyre to correlate with the occurrence of large fire years. We note that a similar feature emerges in the R1 and R3 simulations, while for R2 such negative anomalies are revealed only in the Greenland and Norwegian seas.





For the 5% coldest and driest model-years, the observed ocean circulation pattern of R2 and R3 features the intergyre gyre

in its cyclonic phase, driving positive SST anomalies in the Labrador Sea and the central subpolar North Atlantic. Accordingly, Labrador Sea water propagates southwards, indicating an expanding subpolar gyre. Hence, the polar front in the North Atlantic is displaced southwards, resulting in cold, subpolar water advection into the subtropical region in the Newfoundland Basin (Núñez Riboni et al., 2012).

A positive anomaly in the 500 hPa geopotential height over Greenland and the northern North Atlantic promotes a southward

displacement and weakening of westerlies, and anomalous circulation over Scandinavia with intrusions of dry and cold Arctic air, leading to the negative SAT and precipitation anomalies. Warming in the subpolar gyre and cooling over landmasses in turn promotes higher blocking activity in the subpolar North Atlantic (Fig 5c, (Häkkinen et al., 2011)), further diminishing the moisture transport to the northern Scandinavian region from the mid-latitude North Atlantic.

High oceanic heat capacity contributes to creating and sustaining the anomalous winter conditions observed. The atmo-

spheric and oceanic anomalies, though with a weaker magnitude, tend to persist over spring and summer, most likely due to sea-ice ocean feedbacks via positive sea ice extent anomalies in the Barents, Greenland and Baltic sea regions. However, our analyses show that the summers following extreme cold and dry winters are not exceptional in terms of the MDC threshold.

This work is focused on one specific element of the subpolar North Atlantic ocean circulation with potential to drive SST

anomalies, which can be further coupled to the blocking- and forest fire activity over Scandinavia. Any relationship between ocean circulation and forest fire activity is a product of ocean-atmosphere coupling acting in both directions. We find that a single, one-way driver is not sufficient to describe the model droughts observed.

The outlook for obtaining skillful model-based, longer term (annual to multidecadal scale) projections of future forest fire risk in northern Scandinavia may be improved with a better understanding of 1) past forest fire seasonality and 2) carryover

effect of winter variability in the soil moisture on spring and summer drought conditions. We argue that the integration of winter drought effects into the assessments of warm season drought may improve predictions of forest fire hazard which presently relies exclusively on data from spring and summer seasons.





*Code availability.* MATLAB source code is available upon request from the corresponding author.

*Data availability.* MPI-ESM Climate model simulations are available from https://esgf-node.ipsl.upmc.fr/projects/esgf-ipsl/ or upon request
from Johann Jungclaus.

## Appendix A:  List of years with exceptional dry and warm summers (MDC-based) and dry and cold winters (5th percentile)

Note that the years identified as potential large fire years using the thresholds in Sect. 2.2 are not expected to coincide directly
with true large fire years. The years are in the range of the simulation time period AD 850-2005, and the list in Tab. A1
demonstrates the temporal distribution of potential large fire events in the model simulations. Exceptional dry and cold winters
are identified as the 5th percentile of spatially averaged DJF SAT and precipitation, calculated for the 22 model grid cells in
the northern Scandinavian region and every model-year between AD 850-2005.



*Author contributions.* TN and DVD designed the study and performed the analyses. JJ shared data from the MPI-ESM-P model simulations. All authors have discussed and given input on the scientific results. TN drafted the manuscript, with all authors contributing in the writing process until the final version.

*Competing interests.* The authors declare that they have no conflict of interest

*Acknowledgements.* T. N, D. D and A. H. recieved financial support from the Research Council of Norway (grant 260400/E10). I. D. was supported by the Canadian National Research Council Canada through Discovery Grant (grant nr. DDG-2015-00026 to I.D.), Swedish Research Council FORMAS (grant nr. 239-2014-1866 to I.D.), The Swedish Institute funded networks CLIMECO and BalticFire (nr. 10066-2017-13 and nr. 24474/2018 to I.D.), and Belmont Forum Project PREREAL (grant nr. 292-2015-11-30-13-43-09 to I.D.). The study was conducted within the framework of the NordicProxy network, which is supported by the Nordic Forest Research (SNS), and consortium GDRI Cold Forests.
The authors thank Eduardo Moreno-Chamarro for providing additional model-based data series.



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

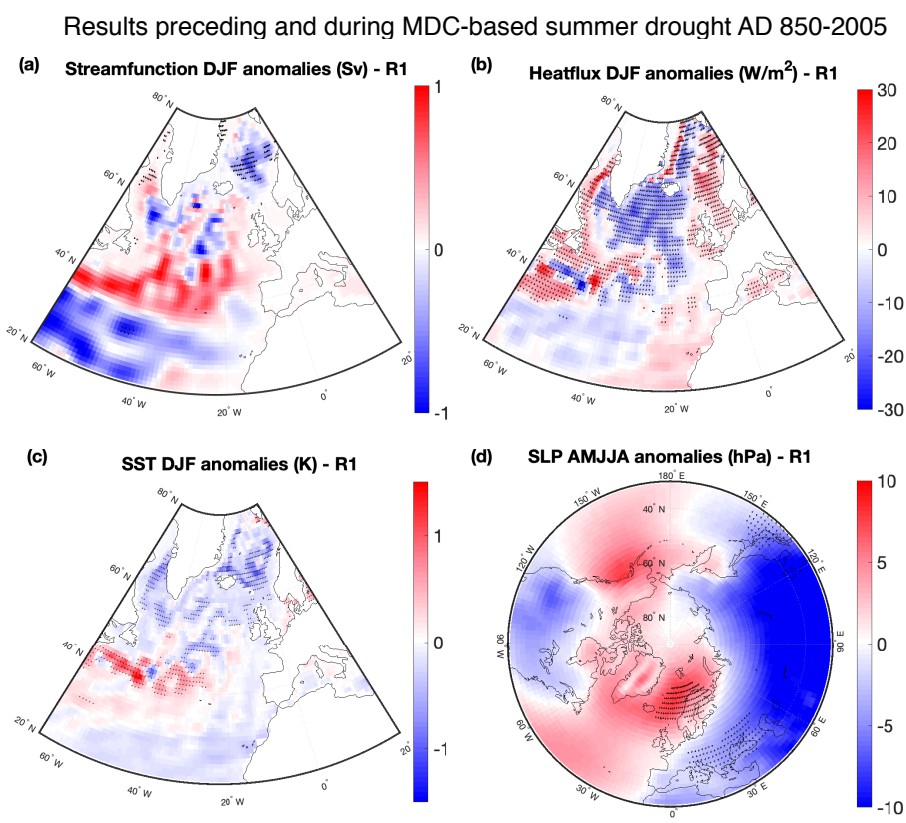

**Figure 1.** R1 composite anomalies related to MDC-based dry summers. Significant anomalies are indicated by black stippling. Panels a-c are for winter (December, January, February: DJF) preceding MDC-based dry summers, panel d is for April, May, June, July, August (AMJJA) the same year as high MDC. (a) Barotropic streamfunction (Sverdrup). (b) Heat flux between ocean and atmosphere (W/m$^2$), positive in the direction from the atmosphere to the ocean. (c) Sea surface temperature (SST) (K), and (d) sea level pressure (hPa).





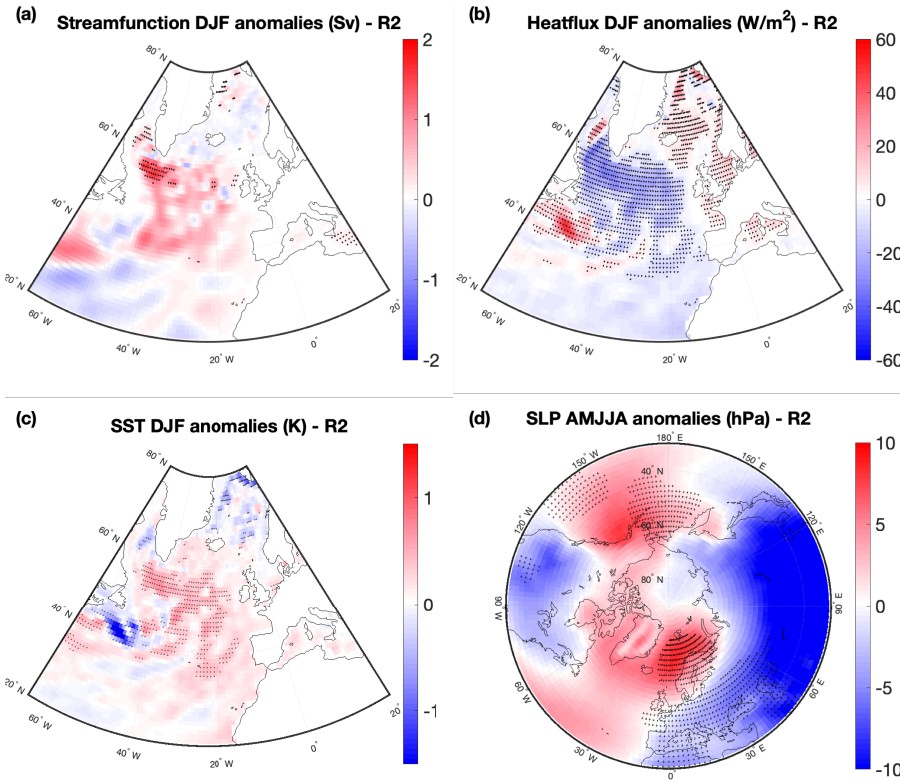

**Figure 2.** R2 composite anomalies related to MDC-based dry summers. Significance and variables in the panels are the same as described in Fig. 1





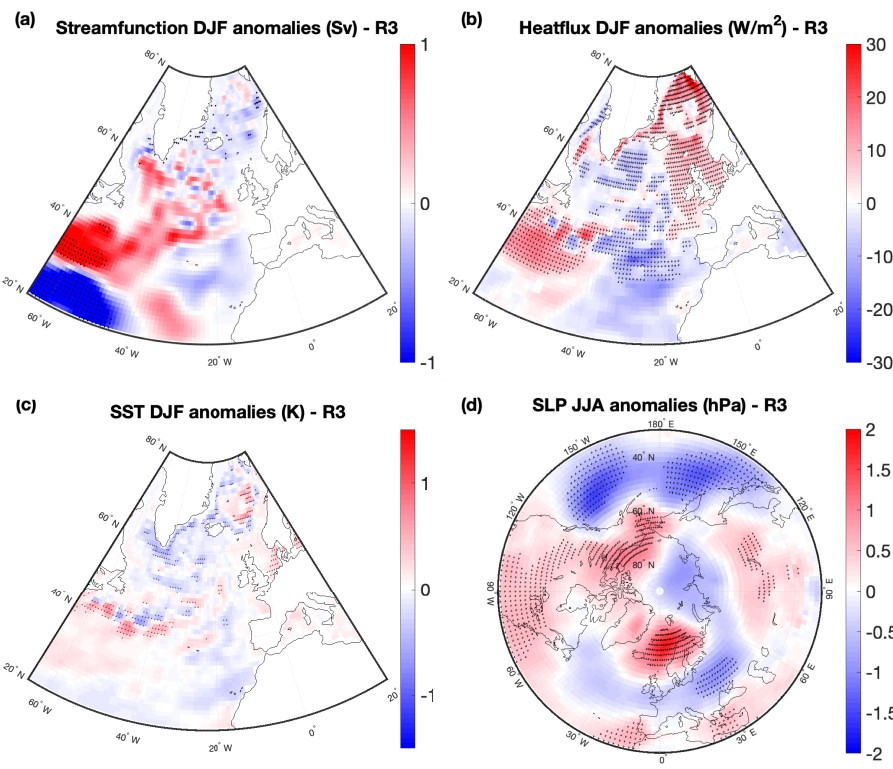

**Figure 3.** R3 composite anomalies related to MDC-based dry summers. Significance and variables in the panels are the same as described in Fig. 1, except panel d for which the time period covered is June-August (JJA).



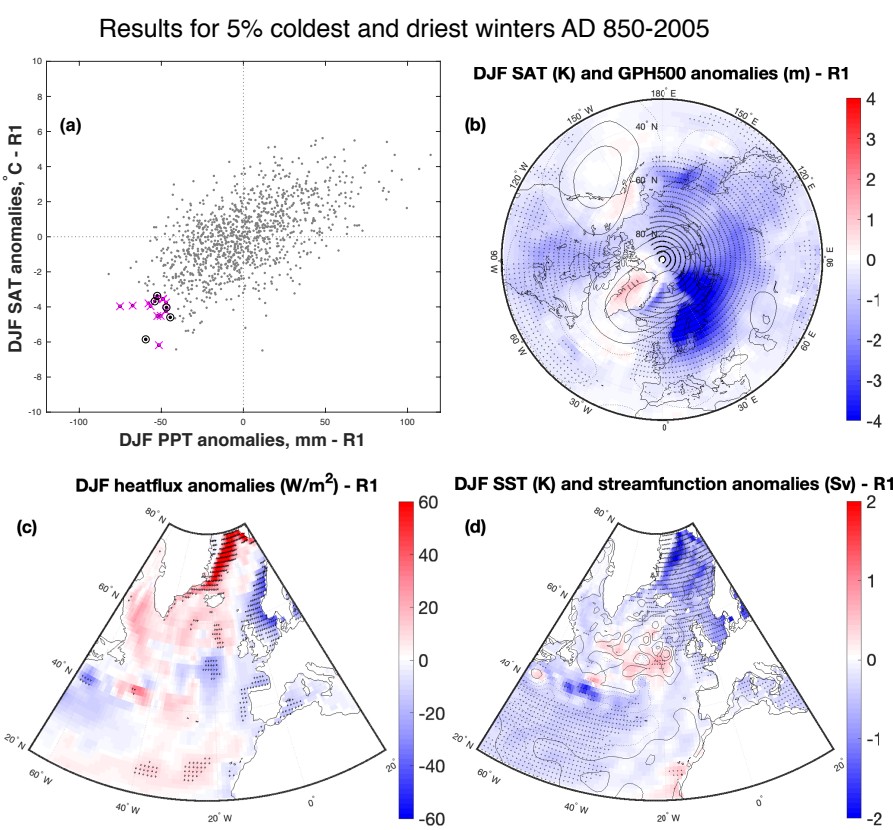

**Figure 4.** R1 composite anomalies for the 5% coldest and driest winters. (a): the spatially averaged surface air temperature (SAT) anomalies versus precipitation anomalies in northern Scandinavia for every year of the period AD 850-2005. The 5% coldest and driest years are highlighted in the leftmost, lower corner of the plot. Magenta denotes years within the fifth percentile associated with weaker subpolar gyre circulation, black denotes years of stronger gyre circulation. (b-d): color maps for composite anomalies with statistical significance indicated by black stippling. Contours are given as solid lines for positive anomalies, dotted for negative anomalies. (b): SAT (K) and 500 hPa geopotential height (contour intervals 10 m). (c): heat flux, (W/m$^2$, positive direction from the atmosphere to the ocean). (d) SST (K) and barotropic streamfunction (Sverdrup, contours at intervals of 0.4 Sv).





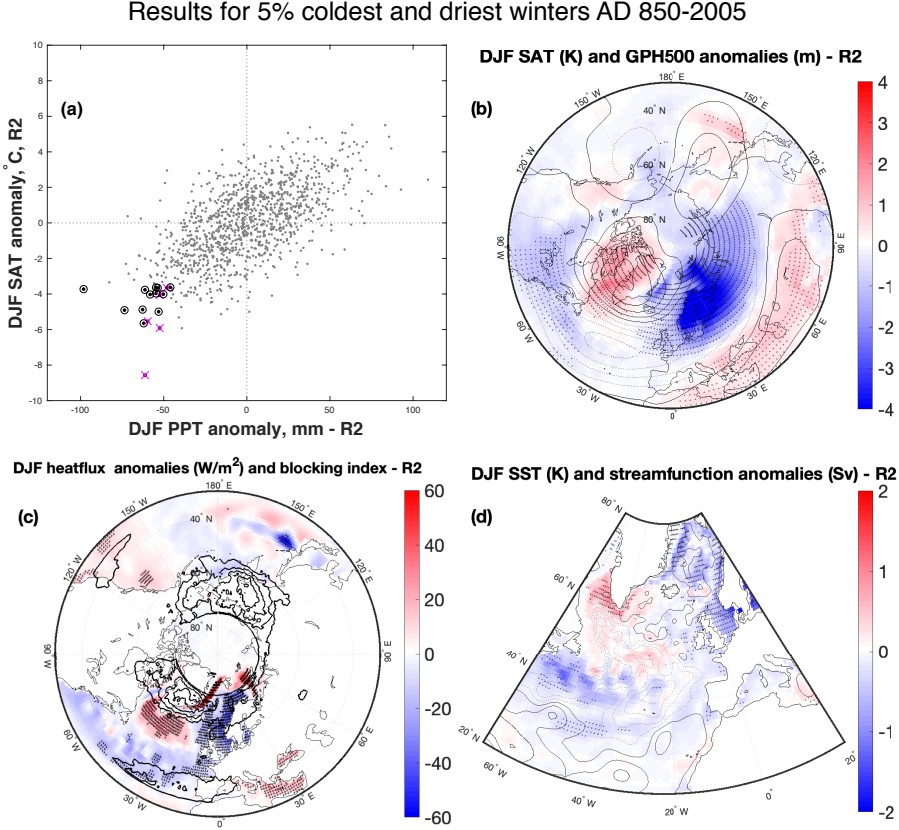

**Figure 5.** R2 composite anomalies for the 5% coldest and driest winters. The information in the subplots is the same as for Fig. 4, except panel (c) also includes the blocking index (contours at 2 units from 2 to 8; one unit in the blocking index corresponds to five or more additional days under blocked atmospheric situations.



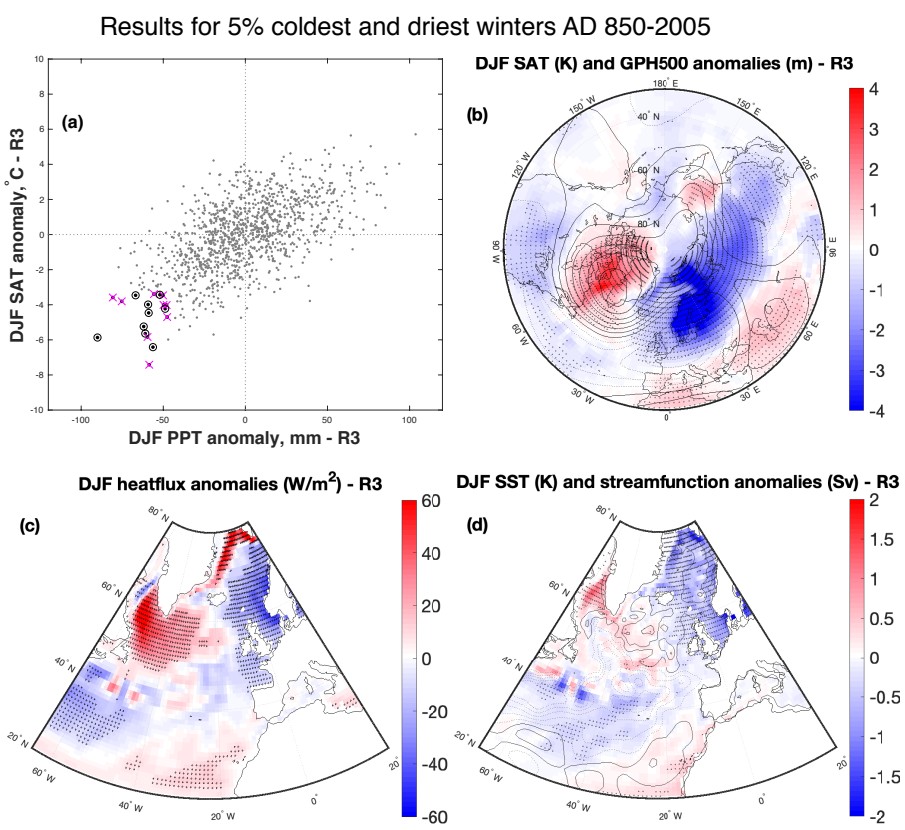

**Figure 6.** R3 composite anomalies for the 5% coldest and driest winters. The information in the subplots is the same as in Fig. 4.





**Table A1.** List of exceptional years for each climate model ensemble member in terms of warm and dry summers (MDC-based) and cold and dry winters (5th percentile)

| Ens. member | total nr. | MDC-based summer drought (year AD) |
|---|---|---|
| R1 | 13 | 876, 959, 967, 1063, 1250, 1292, 40 1421, 1506, 1653, 1744, 1826, 1838, 1882 |
| R2 | 14 | 879, 926, 966, 968, 1081, 1109, 1187, 1200, 1215, 1404, 1434, 1445, 1719, 1779 |
| R3 | 19 | 864, 889, 895, 1102, 1254. 1319, 1408, 1414, 1453, 1458, 1492, 1570, 1599, 1622, 1649, 1704, 1757, 1956, 1983 |
| Ens. member | total nr. | 5th percentile driest and coldest winters (year AD) |
| R1 | 17 | 965, 1021, 1031, 1075, 1242, 1261, 1290, 1451, 1460, 1622, 1624, 1646, 1655, 1675, 1818, 1839, 1944 |
| R2 | 17 | 868, 944, 1261, 1280, 1314, 1317, 1463, 1537, 1610, 1629, 1664, 1687, 1803, 1808, 1872, 1925, 1941 |
| R3 | 18 | 1001, 1106, 1137, 1143, 1277, 1290, 1394, 1477, 1524, 1605, 1613, 1620, 1660, 1717, 1746, 1753, 1771, 1877 |