# Peer review of "Coupling between the North Atlantic subpolar gyre vigor and forest fire activity in northern Scandinavia"

_Climate of the Past, 2019_

## Referee Comment (RC1) · Anonymous Referee #1 · 25 Nov 2019

In general, I found the paper of Nilsen et al. straightforward and neat. My only concern is that I missed the use of paleofire and paleoclimate reconstructions in this study. Observational data retrieved from reconstructions could help to understand the results (or lack of results) of this paper. I encourage the authors to include, at least in the discussion, a paragraph referring to previous paleoclimate and/or paleofire studies carried out in the study area (if there is any), summarizing their main results and comparing them with the simulations. All in all, I recommend the paper for publications with minor suggestions. See below. L 68 and Suppl. Info. Please indicate the differences between R1, R2 and R3 simulations. L86-91 I would recommend to move this paragraph to the previous section (2.1. Data). More details about the MDC are welcome. What data

was used to obtain the monthly drought code? Is that code based on simulations or paleoclimate data?

---

## Referee Comment (RC2) · Anonymous Referee #2 · 12 Dec 2019

General comments ——————

The manuscript by Nilsen et al. is proposing to evaluate to what extent variations in the North Atlantic subpolar gyre might influence atmospheric conditions over northern Scandinavia and contribute to drought and associated risk in forest fire. For this purpose, the authors based their work on a three-member ensemble of last millennium simulations from MPI-ESM climate model. They focus both on summer and winter droughts, since both of them can be of importance for the onset of fire in this region as suggested by recent observations. They do not find any statistically robust linkages between extreme drought and vigor of the subpolar gyre (SPG), leading to reject the

hypothesis they were testing, based on these model simulations. On the opposite, they suggest more robust linkages of droughts with winter NAO and little ice age conditions.

This is an interesting study and hypothesis to be tested. Nevertheless, it was pretty clear that interannual drought extremes (which is what the authors are looking at in the end) would not be necessarily related with SPG variations, since very few literature is proposing such a link. On the opposite, variability in the atmospheric weather regimes was clearly a more natural candidate, so that it is a bit surprising to spend so much time in the manuscript on a hypothesis that was clearly not straightforward. While I am fine with papers showing "negative" results, they need to be well-substantiated by former literature stating the existence of a link which is maybe not robust. Here, this is not really this approach, since the authors seem to choose arbitrarily a hypothesis; they show it does not work, but they do not try an alternative hypothesis to end up with a story that really brings sufficient new scientific knowledge.

This caveat is very well illustrated by the fact that the paper which is proposing a model analysis, with potentially a lot of analysis given the number of fields available, is in the end proposing only two kinds of figures (Fig. 2 and 3 being just the replicate of Fig. 1, and the same for Fig. 5 and 6, replicate of Fig. 4). In fact, Fig. 1-3 and 4-6 can be clearly combined into two figures. Indeed, considering all the three members together can be done within the same analysis of extreme, since they are supposed to reproduce different occurrences of the same climate conditions (only differing by their internal variability, i.e. noise and extreme). Indeed, for leading an analysis of extreme events it is important to consider a large enough sampling, but there is no reason to separate the three members.

On top of this major point, which prevent publication from my point of view, since the manuscript and the analysis are clearly not mature enough, I have a number of specific comments that are explained below.

–––––––––-

Specific points: ——————

- l. 32: "high predictive potential". Of what? Can you please be more specific?

- L. 37-38: "Weaker circulation…" the statement is only true in winter according to Moreno-Chamaro et al. (2017) at least. It would be interesting to insist on this.

- L. 47: remove "testing" after hypothesis.

- L. 74-78: Since you are focusing on drought, it will be useful to better know how is the land surface hydrology is represented. This could be key for soil moisture content and thus drought conditions.

- L. 86-87. The MDC is a soil moisture index, but apparently it is only based on atmospheric temperature and precipitation, while hydrology within the soil and interaction with vegetation might play a role. I assume soil moisture is a variable from the climate model, so I am wondering why the authors do not directly use this variable, which is more representative of the exact processes at play in the model (e.g. hydrological).

- L. 119-124: Since the author are mainly interested in oceanic impact on the atmosphere, I am wondering why they do not directly use ocean heat transport here, which might combine AMOC and SPG transport, but also Ekman part, and might be more physically enlightening, and offering a better process-based understanding of what is going on in their simulation (i.e. more objective and physical approach).

- L. 132: AMJJA or JJA: why is it changing depending on the member?

- L. 135-139: it would be interesting to also see the response of sea ice in order to provide a more complete description of the conditions associated with the droughts. Other variables might be considered to have a more complete view of the processes at play. In fact, the main hypothesis to contradict is: drought conditions are driven by stochastic interannual variability in the atmosphere. Given what is shown afterwards it is not clear that it does not hold. See next comment.

[Figure]

- L. 163-179: When looking at Fig. 1-3, it appears that the only robust oceanic signal preceding the droughts is the heat flux anomaly. Indeed, SST and streamfunction conditions are really not consistent among the different members. Then, this might suggest that the main processes that play a role (apart from chaotic variability in the atmosphere) is the release of heat by the ocean (whatever the process leading to it, e.g. increase of wind, anomalous SST, etc.). Then, it is unclear if it actually leads to the atmospheric conditions over the North Atlantic leading to drought (it can be just a necessary condition, on top of particular atmospheric circulations), or if it is the combination of the specific atmospheric conditions and anomalous ocean heat fluxes the few months before (and what about during the event?) that leads to the drought conditions. More cross-correlation analyses for instance will be helpful to correctly decipher the processes at play (possible if you leave aside the SPG hypothesis on which it is not clear why you focus). Having a more objective approach would have benefited the study and could have offered a clearer view of the processes at play for drought conditions.

- L. 180: "proposed mechanistic coupling": not clear to me what this refers to. Can you please be more explicit? I feel that there is indeed a lack of mechanistic approach in this study, or at least too superficial.

- Discussion section: I am wondering here why you focus on interannual variability, while 10-year drought conditions might have had stronger linkages with the SPG (e.g. your hypothesis). Is it because such low frequency variability in drought have little influence on fire conditions?

- L. 193: "is likely": This word has a strong meaning in climate science due to IPCC influence. Here I do not feel your demonstration was sufficient to use such a wording since I see no clear demonstration of the subsequent proposed processes, nor supporting references.

- L. 199: "consistent oceanic heat loss": can you specify where?

- Table A1: there is a weird "40" in the second row on the right handside.

---

## Author Comment (AC1) · 9 Jan 2020

**Response to comments by anonymous reviewer #1**

CPD manuscript "Coupling between the North Atlantic subpolar gyre vigor and forest fire activity in northern Scandinavia"

By the authors                                                                                          09.01.2020

We would like to thank the reviewer for a clear and constructive review report. Please find below the point-by-point comments and answers. Important details about the revision are found in the author response to referee report 2.

Reviewer comment: "My only concern is that I missed the use of paleofire and paleoclimate reconstructions in this study. Observational data retrieved from reconstructions could help to understand the results (or lack of results) of this paper. I encourage the authors to include, at least in the discussion, a paragraph referring to previous paleoclimate and/or paleofire studies carried out in the study area (if there is any), summarizing their main results and comparing them with the simulations."

This comment is highly relevant and has been discussed between the authors already. In the revision we will expand the discussion to involve published proxy data analyses in the manuscript, representing multidecadal to centennial scale variability due to the inherent low temporal resolution of the proxy data. Very few publications present proxy records for subpolar gyre circulation strength. Past surface salinity derived from planktonic foraminifera was used for this matter in Thornalley et al. (2009), Moffa-Sanchez et al. (2014) and Moffa-Sanchez & Hall (2017). Thornalley et al. (2018) used sortable silt to infer past near-bottom current flow speeds. All proxy records originate from marine sediment cores and cover the past millennium, but with low temporal resolution corresponding to multidecadal to centennial timescales. The seasonal variability therefore cannot be studied in detail from these records.

Moffa-Sanchez & Hall (2017) argue from proxy-based salinity records that the subpolar gyre circulation strength was weakened during the Little Ice Age, which is consistent with our hypothesis and the results from Drobyshev et al. (2016), namely that Scandinavian forest fire activity increased during this time period. We will include this information in the revised manuscript, and add more details of the relevant findings of selected proxy studies as suggested by the reviewer. Note that we do not include all main results of these proxy-based studies, since they comprise other and interrelated aspects of northern North Atlantic Ocean variability that are not within our focus.
The only study focusing directly on coupling North Atlantic Ocean dynamics with past forest fire activity in Scandinavia is Drobyshev et al. (2016). We feel this article is thoroughly described in the text already, see e.g the introduction lines 29-30 and discussion lines 210-213.

The relationship between North Atlantic SSTs/gyre circulation and atmospheric blocking has been investigated in model studies (introduction lines 37-41). There are no studies using purely observational data for this type of analysis. The closest alternative might be the study of Häkkinen et

al. (2011) which is already cited, they use wind stress curl and 500 hPa geopotential height from atmospheric reanalysis data  to study the relationship between subpolar gyre circulation and atmospheric blocking events during the 20th century.

Reviewer comment: Please indicate the differences between R1, R2 and R3 simulations.

This will be added to section 2.1 - Data.

Initiated from the last year of the MPI-ESM control simulation, a 400-year spin-up period using AD 850 boundary conditions is the basis for the three simulations. They differ by the settings of the ocean state, parameters and initial conditions, but the differences are considered small enough so that all three simulations are probable for the past climate evolution under parameter and forcing uncertainties (Jungclaus et al. 2014).

Reviewer comment: L86-91 I would recommend to move this paragraph to the previous section (2.1. Data). More details about the MDC are welcome.

In the revised manuscript, the climate model soil moisture variable will be used to extract model years subject to extreme drought instead of the MDC. We make this choice based on comments from referee 2, in addition to the weakness that the MDC cannot properly model cold season drought.

Reviewer comment: what data was used to obtain the monthly drought code? Is that code based on simulations or paleoclimate data?

The text in Sect. 2.1 and 2.2 will be rewritten.  The MDC is calculated using monthly maximum SAT and total precipitation data from the model simulations. In the text we attempt to justify our threshold for extreme drought (in terms of spatial extent and MDC values) by comparing the model MDC with real-world MDC and LFY thresholds. However, due to your confusion we understand the differences between the model- and real-world must be made even more explicit.

**References**

Drobyshev, I., Bergeron, Y., De Vernal, A., Moberg, A., Ali, A. A., and Nikasson, M.: Atlantic SSTs control regime shifts in forest fire activity of Northern Scandinavia, Scientific Reports, 6, https://doi.org/10.1038/srep22532, 2016.

Häkkinen, S., Rhines, P. B., and Worthen, D. L.: Atmospheric Blocking and Atlantic Multidecadal Ocean Variability, Science, 334, 655–659, https://doi.org/10.1126/science.1205683, 2011.

Moffa-Sánchez, P., Born, A., Hall, I. R., Thornalley, D. J. R., and Barker, S.: Solar forcing of North Atlantic surface temperature and salinity over the past millennium, Nature Geoscience, 7, 275–278, https://doi.org/10.1038/ngeo2094, 2014.

Moffa-Sánchez, P., Hall, I.R. North Atlantic variability and its links to European climate over the last 3000 years. *Nat Commun* 8, 1726, doi:10.1038/s41467-017-01884-8, 2017.

Thornalley, D., Elderfield, H. & McCave, I. Holocene oscillations in temperature and salinity of the surface subpolar North Atlantic. *Nature* 457, 711–714, doi:10.1038/nature07717, 2009.

Thornalley, D. J. R., Oppo, D. W., Ortega, P., Robson, J. I., Brierley, C. M., Davis, R., Hall, I. R., Moffa-Sánchez, P., Rose, N. L., Spooner, P. T., Yashayaev, I., and Keigwin, L. D.: Anomalously weak Labrador Sea convection and Atlantic overturning during the past 150 years, Nature Communications, 556, 227–230, ttps://doi.org/10.1038/s41586-018-0007-4, 2018.

---

## Author Comment (AC2) · 9 Jan 2020

**Response to comments by anonymous reviewer #2**

**CPD manuscript "Coupling between the North Atlantic subpolar gyre vigor and forest fire activity in northern Scandinavia"**

By the authors                                                                                          09.01.2020

We thank the reviewer for a detailed review. Parts of the reviewer's major concerns can be resolved by clarifying the manuscript objectives. There is a motivation for studying the relationship between the subpolar gyre and the Scandinavian forest fire activity, this is presented first in our reply. We then reply to all general and specific comments, and hope to convince the reviewers and the editor that the manuscript indeed brings important new scientific knowledge. We argue that the controversy on publishing negative results makes it even more important to publish this study in a revised form. Our revised manuscript will be more explicit when motivating the hypothesis, and will include new analyses based on reviewer comments and own suggestions.

One of the motivations for this study and for the larger research project PREREAL in general is to improve forest fire prediction systems in the boreal forests on seasonal to multidecadal timescales. This is a difficult problem, because drought conditions favorable for forest fires are mainly driven by atmospheric blocking events that are partly unpredictable. We aim at achieving better understanding of the climate processes involved in changing the average Scandinavian blocking frequency. Variability in the atmospheric weather regimes is indeed a natural candidate as the reviewer points out, but was originally not a main topic in our study due to the generally weak predictive potential on time scales beyond a few weeks. The manuscript was not meant to be a complete study of the meteorological conditions on weekly timescales prior to forest fire activity.

On the other hand, a self-enforcing mechanism coupling the North Atlantic Ocean circulation and the atmospheric blocking conditions would introduce an element of inertia, or memory that could potentially improve future forest fire prediction models in the boreal region on the desired timescales. The North Atlantic subpolar gyre, the Atlantic meridional overturning circulation (AMOC), or the Atlantic multidecadal variability (AMV) are all candidates for such ocean-atmosphere coupling.

In a simplified manner, we can think of the atmospheric and oceanic potential drivers to Scandinavian drought conditions as respectively, the "atmospheric highway", being the direct route dominated by stochastic variability, and the "oceanic detour" involving an inertial contribution from the large oceanic heat content, in addition to nonlinear interaction with atmospheric stochasticity. The latter is the topic of our study, even though we acknowledge that atmospheric variability may indeed play the dominant role. The physical mechanisms involved in this ocean-atmosphere coupling are generally not well understood, exemplified by Häkkinen et al 2011, stating "Our analysis cannot separate cause and effect between high blocking activity and warm ocean surface, but the existing theory of the midlatitude atmosphere-ocean interaction supports increased persistence of atmospheric anomalies that created oceanic anomalies in the first place (ref)."

Our intention was not to formulate a complicated hypothesis, and we can ensure the reviewer that it is not arbitrary. The ocean circulation of the subpolar gyre stood out as a prominent candidate to influence the Scandinavian drought and forest fire activity considering Drobyshev et al. (2016). Figure 2 of their paper shows correlation maps between annual Scandinavian forest fire activity and monthly SST in the northern North Atlantic, with maximum correlation occurring in the subpolar gyre in April and May. We chose to

focus on subannual timescales because of the fingerprint in seasonal lagged correlation, assuming the majority of the forest fires occurred during summer.

The literature is divided on the matter of the predictive potential of the subpolar gyre on interannual timescales (see also the specific comment and the reply below). In a review paper by Kushnir et al. 2002 it is stated: "At the same time, a hypothesis emerged that extratropical SST anomalies imprint their large persistence on atmospheric variability and could thus be used for short-range climate prediction (e.g., Namias, 1969; 1972; Namias and Cayan, 1981; Ratcliffe and Murray, 1970; Barnett and Somerville, 1983). However, determining the nature and strength of the oceans back interaction on the atmosphere has remained a challenge, and has been the main reason for the use of GCMs in controlled experiments with prescribed SST forcing."

This statement reveals why our study deserved to be published, since it addresses an unresolved topic that is still not properly covered in the existing literature today. Driver-response systems are scale-dependent in space and time, and our individual study is therefore of value despite the negative result.

Based on selected model-based studies on the connection between the subpolar gyre strength and the atmospheric blocking frequency (Häkkinen et al. 2011, Moreno-Chamarro et al. 2017a,b), we are aware that the link has been demonstrated predominantly in winter, on multidecadal timescales and longer. However, even though the interannual timescales are not mentioned explicitly when considering this relationship, we should not automatically dismiss such a relationship without further analysis.

A point-by-point list is found below in this document, with reviewer comments in blue fonts, and author replies in black.

**General comment by the reviewer:**

This is an interesting study and hypothesis to be tested. Nevertheless, it was pretty clear that interannual drought extremes (which is what the authors are looking at in the end) would not be necessarily related with SPG variations, since very few literature is proposing such a link. On the opposite, variability in the atmospheric weather regimes was clearly a more natural candidate, so that it is a bit surprising to spend so much time in the manuscript on a hypothesis that was clearly not straightforward. While I am fine with papers showing "negative" results, they need to be well-substantiated by former literature stating the existence of a link which is maybe not robust. Here, this is not really this approach, since the authors seem to choose arbitrarily a hypothesis; they show it does not work, but they do not try an alternative hypothesis to end up with a story that really brings sufficient new scientific knowledge.

This caveat is very well illustrated by the fact that the paper which is proposing a model analysis, with potentially a lot of analysis given the number of fields available, is in the end proposing only two kinds of figures (Fig. 2 and 3 being just the replicate of Fig. 1, and the same for Fig. 5 and 6, replicate of Fig. 4). In fact, Fig. 1-3 and 4-6 can be clearly combined into two figures. Indeed, considering all the three members together can be done within the same analysis of extreme, since they are supposed to reproduce different occurrences of the same climate conditions (only differing by their internal variability, i.e. noise and extreme). Indeed, for leading an analysis of extreme events it is important to consider a large enough sampling, but there is no reason to separate the three members.

Author response:
As discussed above, we argue that the existing literature justifies our choice of hypothesis, and that the hypothesis should be analysed and studied in a proper manner before drawing conclusions about the assumed outcome. We disagree with the reviewers reasoning in the report: "Nevertheless, it was pretty

clear that interannual drought extremes (which is what the authors are looking at in the end) would not be necessarily related with SPG variations, since very few literature is proposing such a link. "
It is a scientific fallacy to equate the lack of literature sources supporting a hypothesis with an already existing negative result. The gyre-blocking relationship may or may not be relevant also on subannual timescales, even if these timescales are not explicitly mentioned in e.g. Häkkinen et al. 2011, Moreno-Chamarro et al. 2017a,b.

Regarding the type of analyses and presentation of results in the revision, we have decided to include analyses of atmospheric weather regimes in the manuscript revision, using k-means cluster analysis. The final reformulation of the hypothesis remains, but we suggest to test both the atmospheric weather influence on Scandinavian drought, and the oceanic influence, keeping the subpolar gyre as the main component of interest.
We prefer to keep the three simulation ensemble members separated when discussing the results, not including them all in the same pool for analysis. This is because the inter-ensemble results are so different, and we will highlight this in the revision to stress the importance of a multi-model or ensemble-based approach. Furthermore, the figures 1-3 and 4-6 can be merged, as suggested by the reviewer.

Another potentially major change we propose is based on a later comment by the reviewer, namely to use the model soil moisture variable to extract years of exceptional drought. We considered this possibility at an earlier point, but rejected it because the MDC apparently performed adequately for our purpose and the method is designed specifically to consider fire weather conditions. However, it is a major problem that the method cannot properly model winter drought or capture the dry conditions observed for Scandinavia during the Little Ice Age. As before we will define a threshold to extract years subject to exceptional drought. The threshold will be made less conservative than at present in order to have more data points for low-frequency analysis of the gyre-blocking relationship, also to comply with later reviewer comments.

**Specific reviewer comments:**

l. 32: "high predictive potential". Of what? Can you please be more specific?

Author response:
- Msadek et al (2010) mainly consider decadal predictability of AMOC, for which a strong AMOC is associated with warming in the North Atlantic and especially strong sea surface temperature (SST) anomalies in the subpolar gyre.
- Årthun et al. (2017) find that SST anomalies along the North Atlantic Current-Norwegian Atlantic Current pathway can be used to predict SAT over Norway, delayed by up to 7 years.
- Brune et al. 2018 studies the predictability of SST and upper ocean heat content in the subpolar gyre using hindcast experiments, three different initialization techniques and different evaluation periods. They find varying predictive skill for the different setups, with maximum predictive skill from one and up to 5-8 years.
- Buckley et al. 2019 finds predictability of SST and upper-ocean heat content on time scales from interannual and up to 4-6 years in the subpolar gyre.
- The reference to Kushnir et al. 2019 will be replaced by Yeager & Robson 2017 who finds high predictive potential of SST/SAT in the subpolar gyre.

Reviewer comment:

L. 37-38: "Weaker circulation..." the statement is only true in winter according to Moreno-Chamaro et al. (2017) at least. It would be interesting to insist on this.

Author response:
Yes, you are correct. In the revised manuscript we will reformulate the text to take this fact better into account. The majority of Scandinavian forest fires occur during summer, but our study highlights the importance of winter drought for subsequent fires during both the cold and warm season. Using the model soil moisture variable to extract years of exceptional drought in the revision will hopefully capture this memory effect better.

Reviewer comment:

L. 47: remove "testing" after hypothesis.

Author response:
This will be changed in the revision.

Reviewer comment:

L. 74-78: Since you are focusing on drought, it will be useful to better know how is the land surface hydrology is represented. This could be key for soil moisture content and thus drought conditions.

Author response:
There are (at least) two possible strategies to indicate soil drought from the model data. Our original strategy has been to estimate the MDC using model output for the variables maximum surface air temperature and total precipitation. Another strategy is to use the model output variable for soil wetness directly. The model land surface hydrology will be described in more detail in the revision.
 See also the reviewer comment below and our response.

Reviewer comment:

L. 86-87. The MDC is a soil moisture index, but apparently it is only based on atmospheric temperature and precipitation, while hydrology within the soil and interaction with vegetation might play a role. I assume soil moisture is a variable from the climate model, so I am wondering why the authors do not directly use this variable, which is more representative of the exact processes at play in the model (e.g. hydrological).

Author response:
Regarding the soil moisture variable of the model simulations - yes, it is correct that this is model output which could be used in addition or instead of the estimated MDC index. In the revision we will use the model variable to extract years of exceptional drought.

It is true that the MDC method only takes maximum surface air temperature (max SAT) and total precipitation as input, but the algorithm is sophisticated and models soil hydrology processes, both evapotranspiration and soil moisture equivalent after rainfall on a detailed level. Specifically, the evapotranspiration is calculated based on monthly maximum SAT, adjusting for the day length of each month and multiplying by the number of days in the month. The moisture equivalent in the soil is calculated by adding the monthly total rainfall after canopy and surface fuel interception to last month's drought code. The MDC is considered reset during winter in the boreal forest due to considerable precipitation, and is normally calculated for the warm season April to October (Girardin & Wotton, 2009).

Reviewer comment:

L. 119-124: Since the author are mainly interested in oceanic impact on the atmosphere, I am wondering why they do not directly use ocean heat transport here, which might combine AMOC and SPG transport,

but also Ekman part, and might be more physically enlightening, and offering a better process-based understanding of what is going on in their simulation (i.e. more objective and physical approach).

Author response:
Repeating the first point in this response, the results of Drobyshev et al. 2016, Moreno-Chamarro et al. 2017a,b and Årthun et al. 2017 indicate the subpolar gyre as a key region with connection to the Scandinavian drought conditions. Moreno-Chamarro et al. 2017a,b showed that weakened LIA subpolar gyre strength was related to increased winter blocking activity over Scandinavia. Årthun et al. 2017 find that this specific region has high predictive potential for Scandinavian SAT with up to 7 years delay, although they do not specifically consider interannual time scales. Given the literature, it was therefore not an initial priority to consider AMOC, AMV or total ocean heat transport as potential drivers for Scandinavian blocking frequency.
The negative results stimulate further studies on the topic, but in the revised manuscript we choose to extend the analyses to include atmospheric variability instead of other oceanic components. We cannot rule out that the AMOC, AMV or meridional oceanic heat transport may influence Scandinavian blocking, but we hope the reviewer can respect that these oceanic elements deserve their own studies in the time to come.

Reviewer comment:

L. 132: AMJJA or JJA: why is it changing depending on the member?

Author response:
It is no problem to change this so that all figures show JJA for consistency, but we thought it was interesting to find such an exceptional long period of five months SLP anomalies, and that this could be highlighted for the R1 and R2 simulations.

Reviewer comment:

L. 135-139: it would be interesting to also see the response of sea ice in order to provide a more complete description of the conditions associated with the droughts. Other variables might be considered to have a more complete view of the processes at play. In fact, the main hypothesis to contradict is: drought conditions are driven by stochastic interannual variability in the atmosphere. Given what is shown afterwards it is not clear that it does not hold. See next comment.

Author response:
The hypothesis suggested here by the reviewer is much more comprehensive than what we intend to cover in our study. It has already been demonstrated by model-studies that a weakened subpolar gyre strength and reduced northwards ocean heat transport resulted in increased sea ice extent in the Barents and Nordic Seas during the LIA (Moreno-Chamrro et al. 2017a,b). The general consensus seems to be that anomalous heat content in the North Atlantic has a predictable impact on Arctic sea ice (Årthun et al. 2017).
For more details see our next reply.

Reviewer comment:

L. 163-179: When looking at Fig. 1-3, it appears that the only robust oceanic signal preceding the droughts is the heat flux anomaly. Indeed, SST and streamfunction conditions are really not consistent among the different members. Then, this might suggest that the main processes that play a role (apart from chaotic

variability in the atmosphere) is the release of heat by the ocean (whatever the process leading to it, e.g. increase of wind, anomalous SST, etc.). Then, it is unclear if it actually leads to the atmospheric conditions over the North Atlantic leading to drought (it can be just a necessary condition, on top of particular atmospheric circulations), or if it is the combination of the specific atmospheric conditions and anomalous ocean heat fluxes the few months before (and what about during the event?) that leads to the drought conditions. More cross-correlation analyses for instance will be helpful to correctly decipher the processes at play (possible if you leave aside the SPG hypothesis on which it is not clear why you focus). Having a more objective approach would have benefited the study and could have offered a clearer view of the processes at play for drought conditions.

Author response:
The reviewer is absolutely right that the observed ocean heat loss might only be a contributing factor on top of the atmospheric conditions leading to Scandinavian blocking activity. By including analysis of atmospheric weather regimes, we hope to better identify the role of the stochastic atmospheric variability. We will also perform cross-correlation analysis between the subpolar gyre streamfunction/heat flux/SST anomalies and the Scandinavian patterns of SAT/precipitation/sea level pressure.

Reviewer comment:

L. 180: "proposed mechanistic coupling": not clear to me what this refers to. Can you please be more explicit? I feel that there is indeed a lack of mechanistic approach in this study, or at least too superficial.

Author response:
We will reformulate the text in the revised manuscript. Indeed, the physical mechanisms of ocean-atmosphere coupling and the implications for North Atlantic atmospheric blocking are poorly understood, not only by us but more generally by the larger scientific community. Additionally, the specific location where blocking activity occurs is of utmost importance for the Scandinavian forest fire activity, increasing the complexity of the problem. A number of existing, relevant studies investigating ocean-atmosphere coupling and associated blocking activity does not refer explicitly to Scandinavian blocking, typically they classify blocking over a larger region such as (Northwestern) Europe or "the Euro-Atlantic region". The boreal region may or may not be included in their domain of interest, and it is often in the outskirts of the study region (e.g. Barriopedro et al. 2006, Folland et al. 2009, Dunn-Sigouin et al. 2013, Davini et al. 2017, Ghosh et al. 2017. The results may therefore not be directly transferable to our study region, since northern Scandinavian climate is also influenced by the variability in the polar region.

Reviewer comment:

Discussion section: I am wondering here why you focus on interannual variability, while 10-year drought conditions might have had stronger linkages with the SPG (e.g. your hypothesis). Is it because such low frequency variability in drought have little influence on fire conditions?

Author response:
One main reason for focusing on interannual variability is that Drobyshev et al. 2016 found significant correlation between annually burned areas in northern Scandinavia and SST anomalies in the subpolar gyre for spring during the same year. Their study comprised the time periods 1948-1975 and 1996-2014. Given the long timescales generally involved with the North Atlantic Ocean circulation systems, it is also important to study decadal and longer timescales as the reviewer suggests.

We have already at an earlier stage analysed the low-pass filtered subpolar gyre index data to focus on centennial timescales and regimes of forest fire activity (Fig. 1). The figure shows the annual and lowpassfiltered SPG index using a moving average of 100 years. Regime shift analysis is used to divide the subpolar gyre index data into regimes (Rodionov 2004, 2006). Orange and blue marks illustrate respectively, the individual years characterized by exceptional summer or winter drought. The length of the regimes is depending on the cut-off length L, here we have used L=150 after testing a number of possible values. The result shows a clear correspondence between weak gyre circulation during the Little Ice Age (LIA) and increased winter drought conditions over Scandinavia. This is consistent with Moreno-Chamarro et al. 2017a,b. However, we also observe strong drought signals during the period from 1050-1200, despite the strengthened gyre circulation index during this period.

In general, there is weak consistency between the average gyre strength and the occurrence of exceptional drought for the full millennial time period. With this in mind, we speculate that the atmospheric conditions may dominantly drive the Scandinavian blocking as suggested by the reviewer, and the observed anomalous subpolar gyre strength is an additional condition that contributes to changes in the blocking activity only under extreme conditions such as the cold LIA.

In the revised manuscript, we will decrease the threshold to extract more years subject to drought, in order to have more data points for analysis. Drought conditions on decadal and longer timescales will then be analysed again in a similar manner as the figure shown here.

Reviewer comment:

L. 193: "is likely": This word has a strong meaning in climate science due to IPCC influence. Here I do not feel your demonstration was sufficient to use such a wording since I see no clear demonstration of the subsequent proposed processes, nor supporting references.

Author response:
The word "likely" in our text is used as a general term, and we did not think it would be problematic or confused with the definition by the Intergovernmental Panel of Climate Change (IPCC). When using this word, we do not refer to the IPCC, and we do not assign a likelihood to the term. Nonetheless, it is unproblematic to change the formulation to comply with the reviewer's critique.

Reviewer comment:

L. 199: "consistent oceanic heat loss": can you specify where?

Author response:
In the subpolar gyre. This will be changed in the revision.

Reviewer comment:

Table A1: there is a weird "40" in the second row on the right handside.

Author response:
The number 40 will be removed, it is a typo.

**Figure 1**

[Figure]

Figure 1: Annual (gray curve) and filtered subpolar gyre (SPG) index (black curve). A 100-year moving average filter was applied to the data. The raw SPG index is divided into regimes marked by horizontal black lines and vertical red bars. Orange markes denote years of exceptional summer drought using the MDC index, blue marks denote years of exceptional winter drought using the 5% coldest and driest winter months.

**References**

Årthun, M., Eldevik, T., Viste, E., Drange, H., Furevik, T., Johnson, H. L., and Keenlyside, N. S.: Skillful prediction of northern climate provided by the ocean, Nature Communications, 8, 15 875 EP –, https://doi.org/10.1038/ncomms15875, 2017.

Barriopedro, D., R. García-Herrera, A.R. Lupo, and E. Hernández.: A Climatology of Northern Hemisphere Blocking. *J. Climate,* **19**, 1042–1063,https://doi.org/10.1175/JCLI3678.1, 2006.

Brune, S., Düsterhus, A., Pohlmann, H., Müller, W. A., and Baehr, J.: Time dependency of the prediction skill for the North Atlantic subpolar gyre in initialized decadal hindcasts, Climate Dynamics, 51, 1947–1970, https://doi.org/10.1007/s00382-017-3991-4, 2018.

Buckley, M. W., DelSole, T., Lozier, M. S., and Li, L.: Predictability of North Atlantic Sea Surface Temperature and Upper-Ocean Heat Content, Journal of Climate, 32, 3005–3023, https://doi.org/10.1175/JCLI-D-18-0509.1, 2019.

Davini, P., Corti, S., D'Andrea, F., Rivie`re, G., & von Hardenberg, J.: Improved Winter European Atmospheric Blocking Frequencies in High-Resolution Global Climate Simulations. Journal of Advances in Modeling Earth Systems, 9, 2615–2634. https://doi.org/10.1002/ 2017MS001082, 2017.

Drobyshev, I., Bergeron, Y., De Vernal, A., Moberg, A., Ali, A. A., and Nikasson, M.: Atlantic SSTs control regime shifts in forest fire activity of Northern Scandinavia, Scientific Reports, 6, https://doi.org/10.1038/srep22532, 2016.

Dunn-Sigouin, E., and Son, S.-W. : Northern Hemisphere blocking frequency and duration in the CMIP5 models, *J. Geophys. Res. Atmos.*, 118, 1179– 1188, doi:10.1002/jgrd.50143, 2013.

Folland, C.K., J. Knight, H.W. Linderholm, D. Fereday, S. Ineson, and J.W. Hurrell.: The Summer North Atlantic Oscillation: Past, Present, and Future. *J. Climate,* **22**,1082–1103, https://doi.org/10.1175/2008JCLI2459.1, 2009.

Ghosh, R., Müller, W.A., Baehr, J. et al. Clim Dyn, 48: 3547. https://doi.org/10.1007/s00382-016-3283-4, 2017.

Girardin, M. P. and Wotton, B. M.: Summer Moisture and Wildfire Risks across Canada, Journal of Applied Meteorology and Climatology, 48, 517–533, https://doi.org/10.1038/s41598-017-07969-0, 2009.

Häkkinen, S., Rhines, P. B., and Worthen, D. L.: Atmospheric Blocking and Atlantic Multidecadal Ocean Variability, Science, 334, 655–659, https://doi.org/10.1126/science.1205683, 2011.

Kushnir, Y., W.A. Robinson, I. Bladé, N.M. Hall, S. Peng, and R. Sutton.: Atmospheric GCM Response to Extratropical SST Anomalies: Synthesis and Evaluation. *J. Climate,* **15**, 2233–2256, https://doi.org/10.1175/1520-0442(2002)015<2233:AGRTES>2.0.CO;2, 2002.

Moreno-Chamarro, E., Zanchettin, D., Lohmann, K. *et al.* An abrupt weakening of the subpolar gyre as trigger of Little Ice Age-type episodes. *Clim Dyn* **48,** 727–744, doi:10.1007/s00382-016-3106-7, 2017a.

Moreno-Chamarro, E., Zanchettin, D., Lohmann, K., Luterbacher, J., and Jungclaus, J. H.: Winter amplification of the European Little Ice Age cooling by the subpolar gyre, Scientific Reports, 7, https://doi.org/10.1038/s41598-017-07969-0, 2017b.

Msadek, R., Dixon, K. W., Delworth, T. L., and Hurlin, W.: Assessing the predictability of the Atlantic meridional overturning circulation and associated fingerprints, Geophysical Research Letters, 37, https://doi.org/10.1029/2010GL044517, 2010.

Rodionov, S. N.: A sequential algorithm for testing climate regime shifts, *Geophys. Res. Lett.*, 31, L09204, doi:10.1029/2004GL019448, 2004.

Rodionov, S. N.: Use of prewhitening in climate regime shift detection, *Geophys. Res. Lett.*, 33, L12707, doi:10.1029/2006GL025904, 2006.

Yeager, S. G., A. R. Karspeck, and G. Danabasoglu: Predicted slowdown in the rate of Atlantic sea ice loss, *Geophys. Res. Lett.*, *42*, 10,704–10,713, doi:10.1002/2015GL065364, 2015.